# Assessment of Satisfaction with Pharmacist-Administered COVID-19 Vaccinations in France: PharmaCoVax

**DOI:** 10.3390/vaccines10030440

**Published:** 2022-03-14

**Authors:** Arthur Piraux, Marie Cavillon, Aline Ramond-Roquin, Sébastien Faure

**Affiliations:** 1University of Angers, Inserm, CNRS, MINT, SFR ICAT, F-49000 Angers, France; sebastien.faure@univ-angers.fr; 2K.Stat Consulting, F-75014 Paris, France; marie.cavillon@kstat-consulting.com; 3University of Angers, Univ Rennes, EHESP, Inserm, IRSET-ESTER, SFR ICAT, F-49000 Angers, France; aline.ramond@univ-angers.fr; 4Département de Médecine Générale, University of Angers, F-49000 Angers, France

**Keywords:** COVID-19 vaccine, vaccination, pharmacist, prevention, primary healthcare

## Abstract

Background: COVID-19 vaccines are among the most effective measures to reduce serious illness and death from infection with the highly contagious SARS-CoV-2 virus. To improve vaccine accessibility, pharmacists in France have been authorized to administer COVID-19 vaccinations since March 2021. This study aims to assess satisfaction among French people receiving their COVID-19 vaccination from a community pharmacist. Methodology: The PharmaCoVax study was conducted in French community pharmacies from 16 March to 30 June 2021. Interested pharmacists completed an online participation form, giving them access to the self-administered questionnaire. People receiving a pharmacist-administered COVID-19 vaccination completed this questionnaire in the pharmacy. Results: Among the 442 pharmacists involved, 123 actively participated in the study. Overall, 5733 completed questionnaires were analyzed. A proportion of 59% (*n* = 3388) of those who received a pharmacist-administered COVID-19 vaccination had previously received their influenza vaccination, most often in the same pharmacy (*n* = 1744). Only 24% (*n* = 1370) of people visiting a pharmacy had tried to obtain their COVID-19 vaccination elsewhere. Satisfaction was excellent with a rating of 4.92 out of 5.00, and the net promoter score was 93. Conclusions: The pharmacist-administered COVID-19 vaccination service was overwhelmingly appreciated by users. The trust placed in pharmacists may explain the desire to have them perform additional vaccinations.

## 1. Introduction

COVID-19 vaccination is one of the most effective measures to reduce serious illness and death from infection with the highly contagious SARS-Cov-2 virus. In France, vaccination was made available in hospitals, specific vaccination centers, and primary healthcare practices [1,2,3,4,5]. By 30 June 2021, half the population in France had been vaccinated (31% were fully vaccinated against COVID-19, and 20% only partly) [6]. In recent years, community pharmacists have been made responsible for an increasing number of medical acts. In 2019, they were authorized to administer influenza vaccinations [7]. Pharmacists had an active, relevant, and effective involvement in these vaccination programs. During the winter 2020 influenza vaccination program, 87% of pharmacies in France participated and more than one in three influenza vaccinations were administered in a pharmacy [8].

Since March 2021, pharmacists have been authorized to prescribe and administer COVID-19 vaccines [9]. Only pharmacists trained for this procedure, during their initial training or continuing professional development, can offer this service. A theoretical training, associated with a practical training to the act of vaccination, was necessary for the validation of this new mission. To increase vaccination provision, both pharmacy students and pharmacy technicians with practical training were able to administer the vaccine, but not prescribe it [10,11].

Considering the increasing role of community pharmacists in vaccination programs, understanding user satisfaction will be valuable if this role is to continue evolving. For this reason, the “PharmaCoVax” survey was launched. The objective of this study is to evaluate the satisfaction of people who received their COVID-19 vaccination from a community pharmacist during the first period of their involvement in the COVID-19 vaccination program.

## 2. Materials and Methods

To evaluate satisfaction, a self-administered questionnaire (PharmaCoVax) was offered to people who received a COVID-19 vaccine in a community pharmacy in France between 16 March and 30 June 2021. Their participation was voluntary and anonymous.

### 2.1. Community Pharmacist Involvement

An email presenting the “PharmaCoVax” study was sent to a large panel of community pharmacists on 16 March 2021. Moreover, the study was promoted through many networks (Appendix A) and social media platforms (Facebook^®^, Twitter^®^, and LinkedIn^®^). Over 90% of French community pharmacists were contacted [12,13].

Only pharmacies offering COVID-19 vaccination were eligible to participate in the study. In March 2021, about 40% of French pharmacies (*n* = 8000) offered this service, increasing to more than 14,000 in December 2021 [14]. Pharmacists interested in participating completed an online form including their name, email address, region, the number of vaccinators in their pharmacy, the pharmacy location (rural/urban/shopping center), and annual turnover (< EUR 1100 k/EUR 1100 k–EUR 2200 k/> EUR 2200 k). These last two elements follow classification usually used in France [15].

When a pharmacist completed the form, an email was sent to them with the “PharmaCoVax” survey (available online on the Microsoft Forms^®^ platform and on paper), the method for returning questionnaires, and the pharmacy identification number. Pharmacists could then return each completed survey by email, online, or by post.

### 2.2. Respondent Recruitment

Every person receiving a COVID-19 vaccination in a participating pharmacy was invited to complete the survey during the 15 min post-vaccination observation period. However, as the respondent profile directly depended upon vaccination eligibility criteria, which was expected to vary according to government authorization, the study period was divided as follows:
From 16 March to 10 April (people over 55 years old, with comorbidities).From 11 April to 30 April (people over 55 years old, with or without comorbidities).From 1 May to 30 May (people over 18 years old, with comorbidities and all people over 55 years old).From 31 May to 30 June (people over 18 years old, with or without comorbidities).


### 2.3. Data Collection

The survey contained 11 questions divided into three main sections (Appendix B and Appendix C). The first section concerned respondent previous vaccination history including influenza vaccination (at least once, in any year) and attempts to obtain the COVID-19 vaccination by other means. The second section concerned the reasons why the respondent chose to be vaccinated in a pharmacy with seven questions related to respondent opinions on pharmacist involvement in the COVID-19 vaccination program. The third section measured overall respondent satisfaction with the pharmacist-administered vaccination (scale from 1–5), and whether they would recommend this service (scale from 1 to 10 with the net promoter score (NPS)).

The NPS is an indicator usually used in industries or shops to measure customer loyalty and service recommendation [16,17]. The simplicity of this indicator enables it to be used in new fields, including health [18]. The respondent is asked a single question about how likely they are to recommend the service to which they can respond on a scale from 1 (not at all likely) to 10 (extremely likely). People who answer 9 or 10 are considered promoters, those answering 7 or 8 are considered passives, and the others are considered detractors. The NPS is the promoter percentage minus the detractor percentage.

### 2.4. Statistical Analysis

Data were described using numbers and percentages (for qualitative variables) or mean and standard deviation (for quantitative variables). In addition to its distribution, the answers to whether the respondent would recommend pharmacist-administered COVID-19 vaccination were analyzed using the NPS.

After these univariate descriptive analyses, comparative bivariate analyses were performed to compare respondent satisfaction according to the survey period and history of influenza vaccination.

Inter-group differences were assessed using Chi-square or Fisher tests for qualitative variables, and nonparametric Wilcoxon tests for quantitative variables. All analyses were performed using SAS software (Statistical Analysis System). A *p*-value of <0.05 was considered significant and *p* < 0.01 was considered very significant. The Cramér’s V statistic, and more precisely its absolute value, was used to measure association between the considered variables.

## 3. Results

### 3.1. Community Pharmacy Characteristics

A total of 442 pharmacists returned forms, and 27.8% (*n* = 123) were finally active in the study (at least one respondent survey returned). Table 1 compares the profile of pharmacies participating in the study with that of pharmacies throughout France. The comparison used national data from the National Institute of Statistics and Economic Studies (INSEE) for the regional distribution [19]. Panel data from the audit and consulting firm KPMG were used for pharmacy location and turnover due to lack of official statistics on these variables [15].

Fewer participating pharmacies (18.7%) had an annual turnover below EUR 1100 k, compared with the national pharmacy population (24.1%), but no significant difference was observed (*p* = 0.402). Participating pharmacies were more likely to be in rural areas (53.7% vs. 36.8%); our sample appears to be significantly different from the reference population (*p* = 0.002). Pharmacies in the overseas territories of France, Corsica and Normandy were not represented in the participating pharmacy sample. Conversely, the Loire, Grand-Est, Occitania, and Hauts-de-France regions were over-represented, with 70% of responses coming from these regions.

### 3.2. Survey Data: Descriptive Analysis

The completion rate for all questions was between 91.9% and 99.8%.

#### 3.2.1. Characteristics of PharmaCoVax Survey Respondents

Of the 5800 responses received, 5733 could be analyzed (Figure 1).

Almost two-thirds of people (59.1%, *n* = 3388) who received their COVID-19 vaccination in a pharmacy had received an influenza vaccination in the past (Table 2). For more than half of these (56.6%, *n* = 1919), the influenza vaccination was received in a community pharmacy. Finally, 9 out of 10 people (90.9%, *n* = 1744) who received their influenza vaccination at a pharmacy were vaccinated at the same pharmacy as their COVID-19 vaccination.

Only a quarter of people (23.9%, *n* = 1370) vaccinated against COVID-19 in a pharmacy had tried to obtain the vaccine at another setting. About 10% (*n* = 603) went to a vaccination center and the same proportion (11.1%, *n* = 638) to a general practitioner (GP). A very small proportion of respondents (2.3%, *n* = 129) had tried both options.

#### 3.2.2. Respondent Opinions Regarding Pharmacist Involvement in the Vaccination Program

Of the seven statements focusing on respondent opinions about pharmacist involvement in the vaccination program, the first six (relating to accessibility, trust, ease of making an appointment, facilities, time spent, and pharmacist competence) received very strong agreement (Figure 2). For each of these statements, more than 80% of respondents chose “strongly agree”, and approximately 98% agreed or strongly agreed.

Nearly three quarters of respondents (74.1%, *n* = 4248) reported that their opinion of pharmacists improved during the pandemic. Despite the lack of comment section, some of the non-respondents (3.1%, *n* = 177) and negative responses (2.1%, *n* = 120) wrote on their questionnaire that they already had a good opinion of pharmacists before the pandemic.

#### 3.2.3. Respondent Satisfaction

Overall satisfaction is rated at 4.92 out of 5, indicating a very high satisfaction level (*n* = 5297). With a promoter percentage of 94% and detractors making up less than 1%, the net promoter score is 93 (*n* = 5409). This means that an overwhelming majority of users recommend pharmacist-administered COVID-19 vaccinations.

### 3.3. Survey Data: Comparative Analysis

This first descriptive approach used cross-analysis of data, or multivariate statistics, to compare respondent satisfaction with reasons for being vaccinated in a pharmacy (prior pharmacist-administered influenza vaccination, unsuccessful attempts in vaccination centers, or with a general practitioner). Responses were analyzed according to the survey period and the characteristics of the pharmacy where respondents had their COVID-19 vaccination.

#### 3.3.1. Respondent Characteristics Depending on Survey Period

Analysis by survey period reveals a moderate but significant heterogeneity in influenza vaccination history (*p* < 0.0001; Cramér’s V = 0.20) (Figure 3). During the first survey period, three-quarters of respondents (76.6%, *n* = 1550) had been vaccinated against influenza. This percentage decreases gradually, reaching 40.2% (*n* = 315) in the last period. This decline is even more apparent for respondents vaccinated against COVID-19 and influenza in the same pharmacy. In the first survey period, of those people who had received a previous influenza vaccination (*n* = 1549), 61.9% (*n* = 959) were vaccinated against these two infections in the same pharmacy. This reduces to just 34.3% (*n* = 108) in the last period.

Attempt to receive the COVID-19 vaccination is less disparate according to the survey period than previous influenza vaccination (Figure 4). The correlation was statistically significant (*p* < 0.0001) but low (Cramér’s V = 0.06).

#### 3.3.2. Respondent Opinions Depending on Survey Period

Respondent opinions were studied for each survey period and significant heterogeneity was observed for each criterion (*p* < 0.0001), except for time spent at the pharmacy (*p* = 0.197) and ease of making an appointment (*p* = 0.029). Nevertheless, the Cramér’s V reveals a very weak correlation in each case, suggesting no clinically relevant difference according to the period (data not shown for this reason).

#### 3.3.3. Respondent Opinion Depending on Previous Influenza Vaccination

Respondent opinion analysis reveals some interesting differences according to previous influenza vaccination for three criteria (Appendix A). Satisfaction related to the time spent at the pharmacy was higher in those who had already received an influenza vaccination (*p* < 0.05), while respondent opinion of pharmacists improved more for people who have never received a pharmacy-administered influenza vaccination (*p* < 0.001). Additionally, respondents who had been previously vaccinated against influenza, were more likely to declare that a pharmacist would be competent to perform other vaccinations such as the tetanus booster, but the association was lower (*p* < 0.05).

#### 3.3.4. Respondent Satisfaction and Likelihood of Recommendation Depending on Survey Period

The overall satisfaction with pharmacist-administered COVID-19 vaccination (Table 3) is almost the same for each survey period (mean score = 4.92 out of 5).

Concerning likelihood of recommendation, the NPS decreases by an average of one point between period 1 and period 4. Nevertheless, this score remains excellent in each survey period, with a mean of 93.3.

## 4. Discussion

Our survey, based on 5733 responses, reveals a detailed picture of user satisfaction after receiving a COVID-19 vaccination in French pharmacies. Nearly 98% of respondents agreed or strongly agreed with the different statements about their experience receiving their pharmacist-administered COVID-19 vaccination. There was a very high level of satisfaction (4.92/5) and the NPS (93) confirmed that people would recommend the service.

Our results are consistent with a previous study on pharmacist-administered COVID-19 vaccinations in Switzerland. This study revealed that 98.7% of respondents would recommend the service and that respondents had a high satisfaction level for all areas of their experience. Furthermore, as in our study, ease of access and perceived trust were two factors respondents strongly agreed with [21].

Interestingly, the pharmacy-administered COVID-19 vaccination satisfaction rate appears similar to that of the influenza vaccination [22,23]. A study in Western Australia revealed that 99.5% of respondents receiving a pharmacist-administered influenza vaccination were satisfied with the service overall [23]. Most respondents in a Canadian study felt pharmacy-administered vaccination was convenient and that the service was better. They also felt that the pharmacy environment was less stressful, and they appreciated the professionalism and knowledge of the pharmacists [24].

The population eligible for COVID-19 vaccination in the first survey period is very similar to the population eligible for the influenza vaccination in community pharmacies in France (people over 65 years old or with chronic disease), explaining why this group has the highest influenza vaccination rate (47.5%) [20]. This result may be explained by the important role pharmacists have played in this particular period [25]. In France, as in the rest of the world, pharmacist responsibilities have evolved very quickly in order to cope with this pandemic and maintain efficient patient care [26]. They have always remained accessible and available throughout the pandemic and are now authorized to renew long-term treatments. This means older people and people with chronic conditions (such as those in survey period 1) may have had more contact with their pharmacists in the last two years, possibly contributing to their improved opinion of them.

In our study, more than 99% of respondents consider the vaccination more accessible when it is available in a pharmacy. It is reasonable to assume that easier access to the vaccine will improve population adherence and thus increase vaccine coverage [27,28,29]. For this reason, including pharmacists in vaccination programs can help remove structural barriers which impact people’s ability to access vaccination services, therefore improving vaccine uptake [30,31].

Pharmacists are well placed to respond to vaccine hesitancy. As with medicines, pharmacists have an important role reassuring people, providing accurate information, and dispelling misinformation [32]. By nature, pharmacists should be involved in preventing disease and promoting vaccination. Physicians remain the primary source of information for parents of young children, far ahead of pharmacists (81.3% and 12.4%, respectively). However, confidence in the information provided by pharmacists about vaccination is very high (almost 80%) but remains lower than physicians (over 95%) [33]. A recent study analyzed pharmacist perceptions of COVID-19 vaccines based on emergency use authorization (EUA) [34]. Significant heterogeneity was observed according to the vaccine approval date (≤1 year or >1 year). About two-thirds of pharmacists were willing to receive a COVID-19 vaccine themselves and recommend it within one year of its approval, increasing to more than three-quarters one year or more after vaccine approval. In fact, pharmacists and physicians both have very good acceptance of the COVID-19 vaccine (88.8% and 92.1%, respectively) as shown in a recent study into the intention of French healthcare workers to get vaccinated against COVID-19 during the first wave of the pandemic [35]. Only physiotherapists rated higher (95.8%), but nurses were more hesitant (64.7%). Despite the increasing workload of pharmacists during the pandemic, more than a hundred of them participated in this study. The participation rate may seem low when compared to the total number of pharmacies offering COVID-19 vaccinations in France. However, the active participation of 123 centers, throughout France, remains a strength in the pandemic context. This participation can be explained by the simplicity and ease of completing the survey. These criteria were imperative to launching the study and is a study strength. It is estimated that the survey can be explained in just a few seconds, and then people could complete it during the fifteen minutes of post-vaccination supervision. The pharmacist then simply returned the completed surveys. One of the major strengths of this study is its timeliness, as it was initiated at the launch of the pharmacy vaccination campaign. The respondent profile varied according to the changes in vaccine eligibility, which impacted the results.

This study had some limitations. Due to the rapidly changing situation and the tight deadline created by the desire to collect responses from the first people receiving pharmacist-administered vaccinations, it was not possible to ask for an opinion from the Institutional Review Board before starting the study. Only ethics committee approval could be obtained. Therefore, the amount of personal information requested was restricted. Additional data, such as respondent age and gender, would have been very valuable for the comparative analysis and to examine possible relationships.

The qualitative nature of this study has inherent bias. Over 90% of pharmacies belong to the two unions participating in this study (FSPF and USPO). However, not all regional health professional unions gave the same support to this study and only four distributed the survey widely. This explains the over-representation of four regions in the sample meaning the study sample is not representative of all pharmacies in France. Voluntary participation in this study may also introduce selection bias. It is likely that pharmacists who felt more engaged with the survey topic were more likely to respond and offer the survey to people. However, it is difficult to evaluate how much this pharmacist selection has influenced our results. Furthermore, it is possible that pharmacists only asked people who had had a good vaccination experience to complete the survey, resulting in potential overestimation of satisfaction.

## 5. Conclusions

Since pharmacists have been able to administer COVID-19 vaccinations, they have vaccinated more people in cities than any other primary health professional (apart from vaccination centers) [36]. In 2021, more than two-thirds of French pharmacies participated in the COVID-19 vaccination program, administering more than 10 million doses during that year [14]. To improve vaccination coverage throughout France, the pharmacist’s role could be extended to other vaccines such as tetanus, diphtheria, polio, and pneumococcus. This solution has been adopted in several European countries in Europe, Canada, and the United States [28,37,38,39].

The “PharmaCoVax” study has shown that users are very satisfied with their pharmacist-administered vaccination. In addition, it seems necessary to conduct a similar study in the general population, not only in pharmacies, to obtain their opinion about pharmacist-administered COVID-19 vaccinations. Such data are needed to reflect on the potential expansion of the pharmacist’s role providing other vaccines in France, as is the case in several other countries, such as in the United Kingdom, Ireland, or Portugal.

## Figures and Tables

**Figure 1 vaccines-10-00440-f001:**
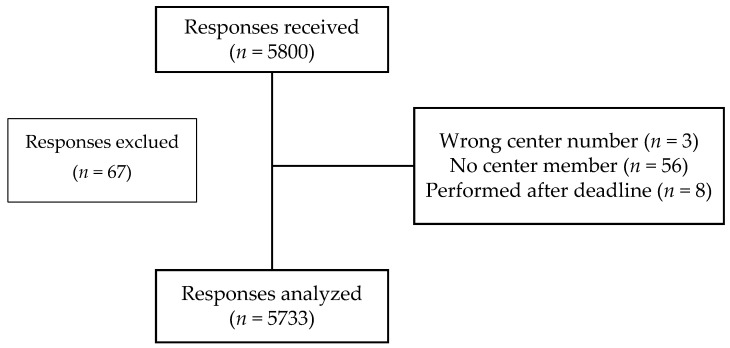
Flow diagram of the PharmaCoVax study.

**Figure 2 vaccines-10-00440-f002:**
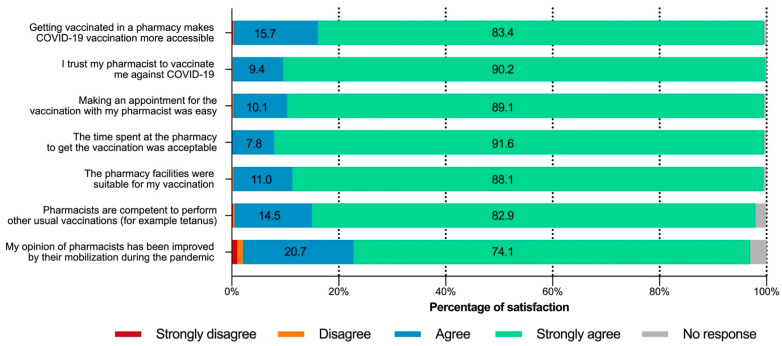
Respondent opinions from the “PharmaCoVax” survey (*n* = 5733).

**Figure 3 vaccines-10-00440-f003:**
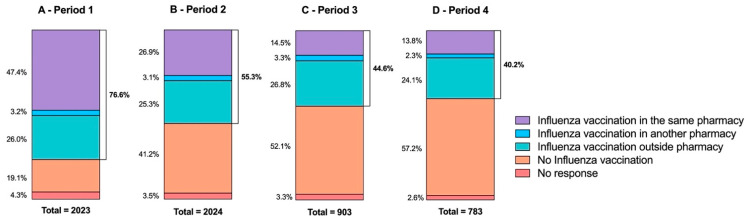
Previous influenza vaccination of respondents depending on survey period (A—period 1, from 16 March to 10 April; B—period 2, from 11 April to 30 April; C—period 3, from 1 May to 30 May; D—period 4 from 31 May to 30 June). Vaccination against influenza is recommended for the most vulnerable individuals (people aged 65 and over, pregnant women, people with certain chronic conditions, and obese people with a body mass index greater than or equal to 40) [20].

**Figure 4 vaccines-10-00440-f004:**
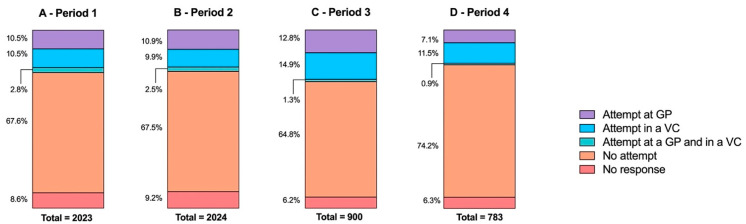
Respondent attempt to receive COVID-19 vaccination depending on survey period (A—period 1, from 16 March to 10 April; B—period 2, from 11 April to 30 April; C—period 3, from 1 May to 30 May; D—period 4 from 31 May to 30 June). GP— general practitioner; VC—vaccination center.

**Table 1 vaccines-10-00440-t001:** Characteristics of community pharmacies participating in the study.

**Region**	**Participating Pharmacy Sample (%)**	**INSEE Data (%) ^1^**
Auvergne-Rhône-Alpes	5 (4.1)	2552 (11.8)
Bourgogne-Franche-Comté	11 (8.9)	975 (4.5)
Brittany	8 (6.5)	1052 (4.9)
Central France—Loire Valley	1 (0.8)	806 (3.7)
Corsica	0 (0.0)	135 (0.6)
Grand-Est	24 (19.5)	1630 (7.6)
Hauts-de-France	22 (17.9)	2020 (9.4)
Ile-de-France	6 (4.9)	3631 (16.8)
Normandy	0 (0.0)	969 (4.5)
New Aquitaine	4 (3.3)	2143 (9.9)
Occitania	18 (14.6)	2037 (9.4)
Loire	23 (18.7)	1123 (5.2)
Provence-Alpes-Côte d’Azur	1 (0.8)	1896 (8.8)
Overseas	0 (0.0)	611 (2.8)
Total	123 (100.0)	21,580 (100.0)
**Pharmacy location**	**Participating pharmacy sample (%)**	**KPMG data (%) ^2^**
Rural	66 (53.6)	226 (36.8)
Urban	52 (42.3)	351 (57.3)
Shopping center	5 (4.1)	37 (5.9)
Total	123 (100.0)	614 (100.0)
**Annual turnover**	**Participating pharmacy sample (%)**	**KPMG data (%) ^2^**
< EUR 1100 k	23 (18.7)	148 (24.0)
EUR 1100–2200 k	68 (55.3)	308 (50.2)
> EUR 2200 k	32 (26.0)	158 (25.8)
Total	123 (100.0)	614 (100.0)
**Number of vaccinators**	**Participating pharmacy sample (%)**	**National data (%)**
1	12 (9.8)	NA
2	60 (48.7)	NA
3	39 (31.7)	NA
4 or more	12 (9.8)	NA
Total	123 (100.0)	NA

^1^ The regional pharmacy distribution was compared with comprehensive data from the National Institute of Statistics and Economic Studies—INSEE [19]. ^2^ Data on pharmacy location and annual turnover were compared with data from a KPMG accounting firm [15].

**Table 2 vaccines-10-00440-t002:** Respondent characteristics concerning influenza vaccination history and COVID-19 vaccination attempts.

**Previous influenza Vaccination ***	**Frequency (%)**
Flu vaccination in the same pharmacy	1744 (30.4)
Flu vaccination in another pharmacy	175 (3.1)
Flu vaccination outside pharmacy	1469 (25.6)
Never received flu vaccination	2138 (37.3)
No response	207 (3.6)
**Attempted COVID-19 vaccination**	**Frequency (%)**
Attempt at a general practitioner (GP)	603 (10.5)
Attempt in a vaccination center (VC)	638 (11.1)
Attempt at a GP and in a VC	129 (2.3)
No attempt	1897 (68.0)
No response	466 (8.1)

* A respondent was considered to have had a previous influenza vaccination if the vaccination had been received at least once, in any year. Vaccination against influenza is recommended for the most vulnerable individuals (people aged 65 and over, pregnant women, people with certain chronic conditions, and obese people with a body mass index greater than or equal to 40) [20].

**Table 3 vaccines-10-00440-t003:** Respondent satisfaction with pharmacist-administered COVID-19 vaccination, using satisfaction score from 1 to 5, and likelihood of recommending the pharmacy vaccination service, using NPS score.

**Satisfaction**	**Sample Size**	**Respondents (%)**	**Mean**	**SD**
Period 1	2023	1880 (90.9)	4.932	0.274
Period 2	2024	1840 (92.4)	4.926	0.275
Period 3	903	834 (94.9)	4.897	0.345
Period 4	783	743 (92.4)	4.891	0.337
All periods	5733	5297 (92.9)	4.918	0.296
**Recommendation**	**Sample size**	**Promotors, %**	**Detractors, %**	**NPS, %**
Period 1	1903	94.5	0.4	94.2
Period 2	1896	94.4	0.6	93.8
Period 3	860	92.8	0.6	92.2
Period 4	750	92.4	1.3	91.1
All periods	5409	93.9	0.6	93.3

## Data Availability

The data sets used and/or analyzed during the present study are available from the first author on reasonable request.

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
