# Peer review of "Assessment of Satisfaction with Pharmacist-Administered COVID-19 Vaccinations in France: PharmaCoVax"

_vaccines, 2022, doi:10.3390/vaccines10030440_

Round 1
Reviewer 1 Report
This 15 page article has a simple conclusion: COVID-19 immunization has been appreciated by users. For such a message the text should be shortened, with some information being placed in Supplementary materials. On the other hand it is not clear how many / what proportion of pharmacies in France vaccinating against COVID-19 were invited / participated, an additional flow diagram may be useful.
30-39 Delete, trivial — everybody knows.
40-44 Suggest to delete as well, as not relevant > Start with a slightly amended sentence 44/45 and lead over directly to 46.
50 87% of pharmacies in France (?) — specify.
52 ... upon what training? Add half or full sentence.
65-76 Consider to include that only in 'Supplementary material'
79 N? (=90% of community pharmacists in France (?)) — specify.
81 N? What proportion of all pharmacies offered COVID-19 vaccines?
95 ... complete the survey during the 15 minute post-vaccination observation period. > Delete 91-93
136 What proportion of pharmacies immunizing against COVID-19 are 442 and even more so 123? Apparently a very small sample?
231-239 and Figure 5: suggest to place in Supplementary materials as rather small differences between Yes/No
260 Any detailed questions on why vaccines were dissatisfied? That would be interesting as well.
273-278 This is obvious from 98-102, that segment can be shortened.
287/294 Accessibility — redundant. Condense the text.
297-318 Suggest to delete, as not a subject of your survey. It is possible to continue straight with 319.
325-327 Suggest a more critical wording about the vaccine hesitancy among pharmacists. If you have comparable data with primary health care physicians in France, add a comparison.
329 High? 123 is what proportion? My guess is that this a very small proportion among all COVID-19 vaccinating pharmacists — or am I wrong? Is that not rather a limitation than a strength?
364 Enthusiastic is a strong term — rather reminds me of propaganda than science — rather suggest 'very satisfied'.
CONTRADICTION? There was no IRB approval (339/340) vs. there was one (375-7) — Any ethical committee?
Author Response
This 15 page article has a simple conclusion: COVID-19 immunization has been appreciated by users. For such a message the text should be shortened, with some information being placed in Supplementary materials. On the other hand it is not clear how many / what proportion of pharmacies in France vaccinating against COVID-19 were invited / participated, an additional flow diagram may be useful.
Firstly, we would like to thank you for your careful analysis of our manuscript and the pertinent comments made.
You will see below that most suggestions have been accepted, allowing us to summarize the information and thus reduce the document length. Two supplementary material documents have been created.
We decided not to present a flowchart showing pharmacy participation of pharmacies since the numbers are indicative and approximate, therefore a flow diagram would be complicated and possibly irrelevant. However, we would like to inform you that some clarifications have been made in the document in response to your suggestions and comments.
Line 30-39: Delete, trivial — everybody knows.
We have followed your recommendation.
Line 40-44: Suggest to delete as well, as not relevant > Start with a slightly amended sentence 44/45 and lead over directly to 46.
We have followed your recommendation and formulated a new introduction. We hope this will be acceptable
Line 50: 87% of pharmacies in France (?) — specify.
We have clarified this point.
Line 52: …upon what training? Add half or full sentence.
We have added a sentence to explain the training.
Line 65-76: Consider to include that only in 'Supplementary material'
We have followed your recommendation
Line 79: N? (=90% of community pharmacists in France (?)) — specify.
We have clarified this point as much as we can. 90% is only approximate as it depends on the network and union. Unfortunately, we don’t have an exact number.
For your information, each union sent the request to participate in the PharmaCoVax study to all its members, but also to all other pharmacists in France. We cannot be sure that every pharmacy has received the information, which is why we have estimated the percentage at 90%. However, the multiplicity of the channels used also favored the dissemination of the information.
Line 81: N? What proportion of all pharmacies offered COVID-19 vaccines?
About two-thirds of French pharmacies offer COVID-19 vaccines (at peak vaccination). There were approximately 8,000 pharmacies at the time of the study. We have added this value in the document.
Line 95: ... complete the survey during the 15 minute post-vaccination observation period. > Delete 91-93
We have accepted your proposal and changed the text accordingly.
Line 136: What proportion of pharmacies immunizing against COVID-19 are 442 and even more so 123? Apparently a very small sample?
We have clarified this information in the methodology section 2.1, following your previous comment (line 81).
Lines 231-239 and Figure 5: suggest to place in Supplementary materials as rather small differences between Yes/Nos
The last criterion is extremely significant; we therefore decided to place Figure 5 in the supplementary results as you suggest and keep the paragraph in the main document. We hope you will find this acceptable.
Line 260: Any detailed questions on why vaccines were dissatisfied? That would be interesting as well.
Unfortunately, we do not have any other explanation for this question. For your information, here are the details of the distribution of the scores. You can see that only 3 people (out of 5733) seem very dissatisfied:
- Score 1: 2 people
- Score 2: 1 people
- Score 3: 19 people
- Score 4: 383 people
- Score 5: 4892 people
- No answer: 436 people
Lines 273-278: This is obvious from 98-102, that segment can be shortened.
We have synthetized this into a single sentence.
Line 287/294: Accessibility — redundant. Condense the text.
We have synthesized the information into a single paragraph.
Line 297-318: Suggest to delete, as not a subject of your survey. It is possible to continue straight with 319.
While this topic is important, it is outside the scope of our research question. We have therefore deleted these two paragraphs.
Lines 325-327: Suggest a more critical wording about the vaccine hesitancy among pharmacists. If you have comparable data with primary health care physicians in France, add a comparison.
We have added more information to this section. A French study evaluated the intention of French healthcare workers to get vaccinated against COVID-19. There was no significant difference between physicians and pharmacists in the acceptance of COVID-19 vaccination. However, nurses were more hesitant about this vaccine.
We also added another source on the confidence of parents of young children in immunization information.
Line 329: High? 123 is what proportion? My guess is that this a very small proportion among all COVID-19 vaccinating pharmacists — or am I wrong? Is that not rather a limitation than a strength?
You are correct in saying that 123 pharmacies represent a small proportion of all those offering the vaccination service (about 1.5% of pharmacies). However, getting 123 investigating centers is a strength, especially in the context of the pandemic. We have chosen to reformulate the sentence.
Line 364: Enthusiastic is a strong term — rather reminds me of propaganda than science — rather suggest 'very satisfied'.
The change was made accordingly.
CONTRADICTION? There was no IRB approval (339/340) vs. there was one (375-7) — Any ethical committee?
We have In France two types of authorizations, depending on the category of study: IRB for studies involving use of personal health data or ethical committee for studies not involving. In our case, we only collected satisfaction data and of the approval of an ethical committee was sufficient.
We modified the document to clarify this point.
Reviewer 2 Report
It is well known that the vaccine is one of most important tool to prevent Covid-19 and to control the pandemic of this disease. To this end, both high coverage and timely vaccination are critical.
This study was aimed to assess the satisfaction with pharmacist-administered Covid-19 vaccination in France and found that this type of vaccination was clearly appreciated by users.
The study design was proper and the manuscript was well written. The limitations were also clearly discussed. The manuscript should be accepted for publication.
There are only two minor points that should be taken when the manuscript will be published:
1. Page 5, line 161, Almost two-thirds of people (59.1%) is not accurate and should be changed to about 60% of people....
2. Page 11, line 328, This sentence is confusing and should be modified.
Author Response
It is well known that the vaccine is one of most important tool to prevent Covid-19 and to control the pandemic of this disease. To this end, both high coverage and timely vaccination are critical.
This study was aimed to assess the satisfaction with pharmacist-administered Covid-19 vaccination in France and found that this type of vaccination was clearly appreciated by users.
The study design was proper and the manuscript was well written. The limitations were also clearly discussed. The manuscript should be accepted for publication.
We would like to thank you for your careful analysis of our manuscript, and the comments made.
You will see below that your suggestions have been accepted.
There are only two minor points that should be taken when the manuscript will be published:
1. Page 5, line 161, Almost two-thirds of people (59.1%) is not accurate and should be changed to about 60% of people....
The change was made.
- Page 11, line 328, This sentence is confusing and should be modified.
We modified the sentence to: “Despite the increasing workload of pharmacists during the pandemic, more than a hundred of them participated in this study”.
Round 2
Reviewer 1 Report
Thank you for having thoroughly revised the manuscript — hope that you share my opinion that it is much improved.
Few minor points:
33ff Review/renumber the sequence of the references, now you have (33) #1-5, (34) #15, (36) #6.
143 Suggest "Almost two thirds of people..."
Author Response
Thank you for your careful review, I totally share your opinion.
- The reference problem has been solved (it was probably related to tracking changes).
- Your suggestion has also been accepted.